# Inflammatory Cytokines and ctDNA Are Biomarkers for Progression in Advanced-Stage Melanoma Patients Receiving Checkpoint Inhibitors

**DOI:** 10.3390/cancers12061414

**Published:** 2020-05-30

**Authors:** Jesper Geert Pedersen, Anne Tranberg Madsen, Kristine Raaby Gammelgaard, Ninna Aggerholm-Pedersen, Boe Sandahl Sørensen, Trine Heide Øllegaard, Martin Roelsgaard Jakobsen

**Affiliations:** 1Department of Biomedicine, Faculty of Health, Aarhus University, 8000 Aarhus, Denmark; jgp@biomed.au.dk (J.G.P.); kraaby@biomed.au.dk (K.R.G.); 2Department of Clinical Biochemistry, Aarhus University Hospital, 8000 Aarhus, Denmark; atm@clin.au.dk (A.T.M.); boesoere@rm.dk (B.S.S.); 3Department of Oncology, Aarhus University Hospital, 8000 Aarhus, Denmark; aggerholm@oncology.au.dk

**Keywords:** melanoma, biomarkers, cytokines, inflammation, immune checkpoint inhibitors

## Abstract

Purpose: Checkpoint inhibitors have significantly improved treatment of metastatic melanoma. However, 40–60% of patients do not respond to therapy, emphasizing the need for better predictive biomarkers for treatment response to immune checkpoint inhibitors. Prorammed death-ligand 1(PD-L1) expression in tumor cells is currently used as a predictive biomarker; however, it lacks specificity. Therefore, it is of utmost importance to identify other novel biomarkers that can predict treatment outcome. Experimental design: We studied a small cohort of 16 patients with advanced-stage melanoma treated with first-line checkpoint inhibitors. Plasma samples were collected prior to treatment initiation and continuously during the first year of treatment. Circulating tumor DNA (ctDNA) level and the expression of ten inflammatory cytokines were analyzed. Results: We found that the ctDNA-level in a blood sample collected after 6–8 weeks of therapy is predictive for response to checkpoint inhibitors. Patients with undetectable ctDNA had significantly longer progression-free survival (PFS) compared with patients with detectable ctDNA (median 26.3 vs. 2.1 months, *p* = 0.006). In parallel, we identified that high levels of the cytokines monocyte chemoattractant protein 1 (MCP1) and tumor necrosis factor α(TNFα) in baseline blood samples were significantly associated with longer PFS compared to low level of these cytokines (median not reached vs. 8.2 months *p* = 0.0008). Conclusions: These findings suggest that the levels of ctDNA, MCP1, and TNFα in baseline and early follow-up samples can predict disease progression in metastatic melanoma patients treated with checkpoint inhibitors. Potentially, these minimally invasive biomarkers may identify responders from non-responders.

## 1. Introduction

Metastatic melanoma is an aggressive cancer for which the incidence rate continues to rise worldwide [1]. Treatment options have been limited for metastatic melanoma patients but with the emergence of novel immunotherapeutic approaches over the past several years, the management of this disease has been vastly transformed. Immunotherapeutic agents targeting programmed cell death protein 1 (PD-1) or cytotoxic T-lymphocyte-associated protein 4 (CTLA-4) on the T cells have demonstrated prolonged overall survival (OS) for advanced-stage patients independent of mutational status [2,3,4,5]. Nonetheless, 40–60% of patients do not respond to immunotherapy and no clearly defined biomarker is available for predicting if patients will benefit from treatment [3,4,6]. The most widely accepted, clinically used predictive biomarker is the expression of PD-1 ligand (PD-L1) on tumor cells, which has been associated with higher likelihood of response to therapy [7]. Yet, up to 41.3% of melanoma patients with PD-L1 negative tumors respond to immunotherapy [4]. Furthermore, PD-L1 expression has been found to display considerable intra- and intertumoral heterogeneity [8], suggesting that PD-L1 expression is not a reliable and specific biomarker. Thus, it remains of utmost importance to uncover better predictive biomarkers for early assessment of response to immunotherapy. This will contribute to the prevention of unnecessary exposure to adverse events related to immunotherapy.

Circulating cell-free tumor DNA (ctDNA) has been extensively studied as a non-invasive biomarker in melanoma and other cancer types. Analysis of ctDNA can identify tumor-specific mutations and has been shown to correlate with tumor burden and clinical outcome [9,10]. Several studies have found that undetectable ctDNA prior to or early after initiating treatment is associated with improved progression-free survival (PFS) and OS [11,12,13,14,15,16]. Furthermore, ctDNA can also be used to uncover mechanisms of acquired resistance at the time of disease progression, allowing therapy to be adapted accordingly [12,16]. Overall, this renders ctDNA a valuable tool for the real-time monitoring of response during treatment. Nonetheless, ctDNA detection can be limited by the sensitivity of detection methods and ‘non-shedding’ tumors.

Immunotherapy is dependent on an activated adaptive immune system, where the focus is to improve the functionality of T cells. However, the activation level of the adaptive immune system is highly dependent on the function of the innate immune system. In this regard, activation of innate immune pathways by damage-associated molecular patterns (DAMPs) is of particular importance for further activation of the adaptive immune system and the generation of an anti-tumor immune response [17,18]. Specifically, DAMP-induced innate immune activation results in the production of inflammatory cytokines that support activation of the adaptive immune system and increase the influx of immune cells into the tumor microenvironment [19,20]. The ability to mount an innate immune response, and thereby also the ability to activate the adaptive immune system, varies between individuals. Here, intrinsic levels of blood inflammatory cytokines may reflect immune activation status of the individual. In this regard, high baseline levels of interferon γ (IFNγ), interleukin 6 (IL-6), and IL-10 have been associated with response to nivolumab [21]. Furthermore, a high level of transforming growth factor β (TGF-β) has been associated with increased response to nivolumab but worse outcome to ipilimumab [22,23]. Despite considerable research within the field, no consensus has been reached so far.

Here, we investigated whether ctDNA and immune-related cytokines can predict treatment outcome in metastatic melanoma patients treated with first-line checkpoint inhibitors. We find that undetectable ctDNA at 6-8 weeks after treatment initiation as well as high baseline levels of the cytokines IFNβ, MCP1, and TNFα are predictors of superior PFS. Collectively, these data suggest that certain cytokines can synergize with ctDNA and function as minimally invasive predictive biomarkers for the effect of checkpoint inhibitor therapy.

### Translational Relevance

Checkpoint inhibitors have significantly increased survival for metastatic melanoma patients. Yet, 40–60% of patients do not respond to checkpoint inhibitor therapy. To provide the best possible treatment, it is of utmost importance to identify responders from non-responders. Currently, there is a lack of well-founded clinically available biomarkers for identification of responders from non-responders. Thus, it is essential to identify new biomarkers for predicting treatment response. Here, we performed an exploratory biomarker analysis in a cohort of advanced-stage melanoma patients treated with checkpoint inhibitors. Through investigation of a broad cytokine panel and ctDNA levels, we demonstrated that plasma levels of MCP1 and TNFα as measured at baseline, and ctDNA levels measured at 6–8 weeks of therapy were strong predictors for treatment response. These findings indicate a potential for using ctDNA, MCP1, and TNFα plasma levels to identify checkpoint inhibitor responders and calls for further investigations.

## 2. Materials and Methods

### 2.1. Patients and Treatment

Patients with unresectable, previously untreated stage III or IV melanoma who received systemic treatment with immune checkpoint inhibitors were eligible for the study. Key inclusion criteria were absence of uveal melanoma, absence of another primary cancer, and no previous diagnosis with cancer. A total of 33 patients were initially enrolled in the study, out of which 17 were later excluded as they did not meet the inclusion criteria: 6 due to earlier treatment; 4 due to other cancer type; 4 receiving first-line BRAF inhibitors, 2 due to lack of baseline samples, and 1 not receiving systemic treatment.

Patients were enrolled between October 2016 and August 2017. All consecutive patients referred to systemic treatment at the Department of Oncology in Aarhus (Denmark) were included. All patients gave informed written consent before inclusion, and the study was approved by the Central Denmark Region Committees on Biomedical Research Ethics (no. 1-10-72-374-15) and performed in accordance with the Declaration of Helsinki.

Patients received pembrolizumab at a dose 2 mg/kg every 3 weeks or nivolumab 1 mg/kg plus ipilimumab 3 mg/kg every 3 weeks, followed by maintenance nivolumab 1 mg/kg.

### 2.2. Disease Characteristics and Response Assessment

Patient demographics and clinicopathologic features included: performance status and metastatic sites at baseline, lactate dehydrogenase (LDH), and any adverse events requiring steroid treatment during one year of treatment. Elevated LDH level was defined as levels above 205 units/liter (U/L) for patients below the age of 70 and above 255 U/L for patients above the age of 70. Tumor biopsies were routinely screened for BRAF^V600E^ mutation status and PD-L1 expression level (</> 1%). Treatment responses were evaluated by Position emission tomography/computed tomography (PET/CT) scans and/or CT of chest, abdomen, and pelvis, and magnetic resonance imaging (MRI) in case of known brain metastases.

### 2.3. Sample Collection and Preparation

Peripheral blood samples (3 × 10 mL Ethylenediamine tetraacetic acid (EDTA) tubes, BD Vacutainer, Plymouth, United Kingdom) were obtained at baseline (immediately before treatment initiation) and every 3–4 weeks during treatment for up to one year after treatment initiation. Plasma was isolated from peripheral blood samples within 2–3 hours after blood collection by 1800× *g* for 10 min at room temperature (RT). Plasma was cryopreserved at −80 °C until analysis.

### 2.4. Cell-Free DNA (cfDNA) Extraction

cfDNA was extracted from 4 mL plasma using the QIAamp Circulating Nucleic Acid Kit (Qiagen, Hilden, Germany) according to the manufacturer’s protocol. The isolated DNA was eluted in 100 μL elution buffer and stored at −80 °C until analysis.

### 2.5. Droplet Digital PCR (ddPCR)

The QX200™ AutoDG™ Droplet Digital™ PCR System (Bio-Rad, Copenhagen, Denmark) was used to perform ddPCR. Samples were analyzed for the following mutations: *BRAF* p.V600E, *NRAS* p.Q61K/p.Q61R, and *TERT* C228T. The *BRAF* and *NRAS* assays were wet-lab validated assays purchased at Bio-Rad. The *TERT* assay was designed and validated in-house. Due to technical limitations with the *TERT* assay, only the C228T mutation was approved for this study. The reaction volume of 22 μL consisted of 2× Supermix for probes (no UTP), 900 nM primers, 250 nM probes, and 5 μL of purified cfDNA for the *BRAF* and *NRAS* assays. For *TERT*, it consisted of 1x Supermix for probes (no UTP), 1200 nM primers, 250 nM probes, 0.5 M betaine, 1 mM EDTA, and 5 μL of purified cfDNA. All reagents were purchased from Bio-Rad. All samples were conducted as duplicates and each run included positive and negative control samples. DNA extracted from the cell lines SK-MEL-28 and T24 was used as positive control for the *BRAF* assay and *TERT* assay, respectively. Genestrands (Eurofins Genomics, Ebersberg, Germany) diluted in cfDNA extracted from blood samples from anonymous donors collected from the blood bank at Aarhus University Hospital were used as positive controls for the *NRAS* assays. The limit of detection (LoD) for each assay was determined using donor cfDNA according to (Milbury biomol detect quant 2014). LoD and assay information can be found in Appendix A. Data analysis was performed using QuantaSoft v.1.7.4.0917 software (Bio-Rad, Copenhagen, Denmark) and reported as copies per ml plasma.

### 2.6. O-Link Analysis

Blood levels of cytokines were analyzed by the proximity extension assay O-link (Immuno-oncology panel, BioXpedia, Aarhus, Denmark). DNA oligo-coupled antibodies targeting 92 different proteins were used in qPCR assay where the number of PCR target fragments were proportional to the concentration of target protein in the input sample. Normalized Protein expression (NPX) units were calculated from the Ct-values obtained from the qPCR run, depictured on a log2 scale. The following cytokines were excluded from the analysis due to undetectable levels in the input samples; FGF2, IL1α, IL2, IL5, IL21, IL33, IL35, and TNFα.

### 2.7. Cytokine Analysis by Meso Scale Discovery

A customized, high-sensitive 10-cytokine U-plex panel (Meso Scale Discovery, Rockville, MMD, USA) was used to analyze plasma levels of IFNβ, IFNγ, IL10, IL1β, IL21, IL6, IL8, IP10, MCP1, and TNFα in cryopreserved plasma samples according to the manufacturer’s protocol. Data were acquired using a QuickPlex SQ 120 instrument (Meso Scale Discovery, Rockville, MD, USA). The lower limit of quantification of each cytokine can be found in Appendix A.

### 2.8. Statistical Analysis

Correlation between ctDNA and LDH or number of metastatic lesions was performed using Spearman’s rank correlation coefficient. ctDNA levels were dichotomized according to the median concentration and an unpaired *t*-test was used to assess the association between ctDNA level (log-transformed) and progression. Fisher’s exact tests was used to evaluate the distribution of patients according to biomarker status (ctDNA, cytokine score, and LDH). Patients with both *BRAF* p. V600E negative tumors and undetectable baseline ctDNA were excluded from all analysis on ctDNA.

Time-to-event analysis were reported using PFS by the Kaplan–Meier method. PFS was defined as time from treatment initiation to the date of first reported progression or death due to any cause. Patients without disease progression or who were still alive at last follow-up were censored at the last follow-up date (2nd of October 2019). Log rank test was performed to assess differences in survival. The assumption of proportional hazard was tested by visualization of Kaplan–Meier plots before any statistic tests were performed. Univariate cox proportional hazards regression was calculated to estimate hazard ratios (HR) for PFS if the assumption was met. Each variable in the univariate cox proportional regression models was tested for proportionality. *p*-values less than 0.05 were considered significant. Analyses were carried out using StataIC version 15.1 (Stata Nordic, Stockholm, Sweden) and Graphpad Prism 8 (version 8.2.0).

## 3. Results

### 3.1. Patient Characteristics

A total of 16 patients with unresectable stage III or IV melanoma and treated with first-line checkpoint inhibitors were enrolled at the Department of Oncology at Aarhus University Hospital. One out of 16 patients (6%) was diagnosed with stage III melanoma, while 15/16 (94%) patients were diagnosed with stage IV melanoma. Eleven patients received treatment with pembrolizumab and five patients received treatment with ipilimumab and nivolumab (Table 1). At the time of study evaluation, the median follow-up was 26 months (range 6.3–35.6 months).

Tumor PD-L1 expression was analyzed on tumor-cells in 11/16 (68.7%) of the patients. Among these 11 patients, five (45.5%) patients displayed PD-L1 expression ≥1% (Table 1). The *BRAF* p.V600E mutation was detected in the tumor biopsy for 10 out of the 16 patients (Table 1). At database lock, nine (56%) patients had experienced disease progression and six (37.5%) patients had died (Table 1), out of which one patient on first-line ipilimumab and nivolumab died of a non-cancer related or treatment-related event. Five patients displayed elevated LDH levels measured at baseline, but this was not associated with worse PFS (*p* = 0.36) (Appendix A).

### 3.2. Detection of ctDNA at Baseline

cfDNA was analyzed for *BRAF* p.V600E, *NRAS* p.Q61K/p.Q61R, and *TERT* C228T mutations, as these are frequently found in melanoma patients [24,25,26,27]. ctDNA was detected by ddPCR at baseline in 9/16 patients (56%). The *BRAF* p.V600E mutations initially identified in the primary tumor biopsy were confirmed in ctDNA for 7/10 patients (70%). It should be noted that for the three patients with *BRAF* p.V600E mutated tumors, but undetectable ctDNA, the only metastatic sites were the brain, lung, and subcutaneous tissue, which are sites known to shed less ctDNA [12,28,29,30]. Mutations in *NRAS* were detected in two patients with one harboring *NRAS* p.Q61K and the other *NRAS* p.Q61R. One patient with a *BRAF* p.V600E mutation had a concurrent *TERT* C228T mutation. For four patients, mutations were identified in neither tumor biopsy nor ctDNA, and these patients were excluded from further analysis. An overview of the identified mutations is shown in Figure 1A. For the nine patients with detectable ctDNA at baseline, the median mutant allele concentration was 133.3 copies/mL plasma (range 10.20–3388) and the median allele frequency was 3.5% (range 0.60–24.6%). Patients who experienced disease progression did not have significantly higher baseline mutant allele concentration than patients without disease progression (*p* = 0.22). Furthermore, higher mutant allele concentrations (defined as above the median concentration) were not associated with PFS (median not reached vs. 6.35 months, *p* = 0.25) and neither was the presence of ctDNA at baseline. However, the baseline mutant allele concentration was significantly correlated with the number of metastatic sites (Spearman *r* = 0.76, *p* = 0.006) (Figure 1B).

### 3.3. Early Changes in ctDNA Level Are Associated with Progression-Free Survival

To assess whether early changes in mutant allele concentration were associated with PFS, we analyzed blood samples longitudinally collected during therapy. A total of 130 plasma samples were available from the 12 patients (median 9 samples per patient, range 4–19) with detectable mutations in tumor or in the baseline blood sample, and ctDNA was identified in 36 (28%) out of the 130 samples. The mutant allele concentration decreased within 2 months upon treatment initiation for the majority of patients, and it remained undetectable in particular for patients who did not experience disease progression (Figure 2A). Next, we divided the patients into two distinct subgroups according to the ctDNA level measured at week 6–8 after treatment initiation. One subgroup consisted of patients who had undetectable ctDNA at week 6–8 regardless of their baseline ctDNA status (*n* = 8), and the second subgroup consisted of patients with detectable ctDNA at week 6–8 (*n* = 4). Patients with undetectable ctDNA at week 6–8 had a significantly longer PFS (median 26.3 months) than patients with detectable ctDNA (median 2.1 months, *p* = 0.006) (Figure 2B). In a univariate Cox analysis, the presence of ctDNA at week 6–8 was a predictor of shorter PFS with a hazard ratio (HR) of 7.89 (95% confidence interval (CI): 1.40–44.6, *p* = 0.019).

### 3.4. ctDNA Levels Mirror Clinical Status during Longitudinal Monitoring

Examples of longitudinal ctDNA monitoring for different clinical situations are shown in Figure 2C. We observed for all patients, who experienced disease progression within one year, that the mutant allele concentration rose either prior to or at progression with a mean lead time of 85 days (range 0–192 days, *n* = 5). The four patients without disease progression had no detectable ctDNA within 6 weeks following treatment initiation and ctDNA remained undetectable in all following samples, except in a single sample for one patient where a diminutive level of ctDNA was detected.

### 3.5. Initial Characterization of Cytokines as Biomarkers for Treatment Response

To explore whether other biomarkers could be used for predicting disease progression, especially in patients not applicable for ctDNA monitoring, we evaluated a large group of blood circulating cytokines. Various cytokines are known to shape and control the immune response and have been associated with anti-tumoral responses. To construct a possible biomarker cytokine panel of potential interest for the patients enrolled on checkpoint inhibitors, we first performed a broad screening of 92 immuno-oncology related proteins using the O-link technology. The O-link panel was validated on plasma samples from a small group of patients who received first-line pembrolizumab treatment. The analysis was conducted on both baseline and follow-up samples to identify baseline levels and discrepancy of cytokines over time. The screening demonstrated large plasma level variation among a large range of different cytokines (Figure 3A). To pinpoint cytokines that could potentially be used as predictive biomarkers, we next searched for cytokines that displayed large inter- or intrapatient variation. In-depth analysis of the cytokine concentrations revealed considerable interpatient variation for a number of the cytokines, including Interferon γ-induced protein 10 (IP-10), IL-8, IL-6, and MCP1 (Figure 3B). In addition to the cytokine screening, we searched the literature for cytokines of potential interest that either reflected innate immune activation or cytokines associated with anti-tumor immune responses [21,31,32,33,34]. Based on these findings, a panel of 10 cytokines (IFNβ, IFNγ, IL-1β, IL6, IL8, IL10, IL21, MCP1, and TNFα) was established for further validation in the patient cohort.

### 3.6. Baseline Levels of MCP1 and TNFα Correlate with Progression-Free Survival

For eight out of ten cytokines (IFNβ, IFNγ, IL6, IL8, IL10, MCP1, and TNFα) in the biomarker panel, we dichotomized the patients into a high and a low cytokine expression group using the median cytokine expression value determined in the baseline sample. For IL21, 9/16 (56%) patients had undetectable levels, and therefore patients were dichotomized according to undetectable (low) and detectable (high) IL21. IL1β was undetectable in the baseline samples in 13/16 (81%) patients and was excluded from further analysis. The interpatient variation for each cytokine is shown in Appendix A. Based on the dichotomization, we then analyzed whether any of the baseline cytokine levels were associated with PFS (Figure 4A and Appendix A). We found that both IFNβ, TNFα, and MCP1 demonstrated a significant difference in PFS between the low and high cytokine group (Figure 4A). The remaining six cytokines analyzed did not demonstrate any difference between low and high cytokine groups (Appendix A). Cox regression analysis revealed that the HR for PFS for patients having high versus low expression levels was 0.081 for TNFα (95% CI: 0.0098–0.66, *p* = 0.019), 0.073 for MCP1 (95% CI: 0.0088–0.61, *p* = 0.016), and 0.22 for IFNβ (95% CI: 0.044–1.09, *p* = 0.063) (Figure 4A). The median PFS was: 8.2 months for low IFNβ and not reached for high IFNβ; 4.2 months for low MCP1 and not reached for high MCP1; 4.2 months for low TNFα, and not reached for high TNFα.

### 3.7. The Immune Cytokine Score Strengthens the Prediction of PFS

Based on the cytokine profiles evaluated above, we next speculated whether an “immune cytokine score” comprised of multiple cytokines could be used to strengthen the association to PFS. MCP1 and TNFα baseline levels provided the best stratification of patients with regard to PFS. Thus, patients having high baseline levels of both MCP1 and TNFα were assigned a cytokine score of 2, while patients having high baseline levels of either MCP1 or TNFα were assigned a cytokine score of 1. Finally, patients having low baseline levels of both MCP1 and TNFα were assigned a cytokine score of 0. Using this immune cytokine score, patients were then dichotomized into a low and a high cytokine score group. Patients with a cytokine score of 0 or 1 were assigned to the low group (*n* = 9), and patients with a cytokine score of 2 were assigned to the high group (*n* = 7) (Figure 4B). Consistent with the results for the individual cytokines, we found that patients with a high cytokine score demonstrated a significantly longer PFS compared with patients with a low cytokine score (log rank test: *p* = 0.0008) (Figure 4B). Median PFS was not reached for the high cytokine score group but was 8.2 months for the low cytokine score group. In contrast to PFS data on individual cytokines, only one patient classified with a high immuno cytokine score experienced disease progression during the follow-up period. Similar to ctDNA, we also evaluated the cytokine score at 6-8 weeks and found that this cytokine score also predicted longer PFS (Figure 4C). Taken together, high levels of MCP1 and TNFα measured at baseline and 6–8 weeks post treatment initiation were predictors of superior PFS in patients treated with checkpoint inhibitors.

### 3.8. ctDNA and Cytokines Are Independent Predictors of PFS

We next investigated if there was an association between ctDNA detection and the cytokine score in terms of defining patients with a favorable biomarker status (undetectable ctDNA and high cytokine score). As both ctDNA and cytokine score were expressed as binary variables, we applied Fisher’s exact test to test for correlation between ctDNA and cytokine score. Among the 12 patients in this study, we did not observe an association between undetectable ctDNA and high cytokine score (*p* = 0.0808) (Appendix A), suggesting that there is a discrepancy between patients identified with undetectable ctDNA and patients identified with a high cytokine score. Moreover, we tested if there were association between ctDNA (undetectable vs. detectable) and LDH (normal vs. elevated) (*p* = 0.55) or between cytokine score (low vs. high) and LDH (normal vs. elevated) (*p* = 0.99); but neither ctDNA nor cytokine score were significantly associated with LDH level (Appendix A). These results suggest that ctDNA detection and the cytokine score are independent predictors of outcome and can thus supplement each other.

## 4. Discussion

The present study investigated the association between PFS and the levels of ctDNA and cytokines in baseline and early follow-up samples from melanoma patients treated with first-line checkpoint inhibitors. We identified a favorable ctDNA profile, defined as undetectable ctDNA levels after 6–8 weeks of therapy, as a good predictor of prolonged PFS. Furthermore, we demonstrated that high baseline levels of the cytokines MCP1 and TNFα predicted longer PFS.

There is a general focus in the field on identifying biomarkers that can predict treatment response to checkpoint inhibitor thereby improving personalized medicine. Baseline LDH level has been accepted as a strong prognostic marker in melanoma [35,36,37,38] and tumor PD-L1 expression has also been suggested as a biomarker for response to anti-PD-1 therapy [39]. However, it has previously been shown that many patients benefit from checkpoint inhibitor therapy, despite their tumors being classified as PD-L1 negative, emphasizing the continued need for better and more reliable biomarkers [4,40].

Here, we investigated the potential of using ctDNA as a predictive biomarker. We observed that undetectable ctDNA at 6–8 weeks of therapy predicted longer PFS, verifying that ctDNA can be a valuable biomarker in melanoma patients receiving checkpoint inhibitors. This is in agreement with other studies showing that ctDNA is a predictor for PFS in melanoma patients treated with checkpoint inhibitors but also targeted therapy [12,13,15,16,41]. We did not find an association between ctDNA detection at baseline and PFS, a finding that is both in agreement with [13] and in contrast to findings of other studies [12,14]. We furthermore observed that monitoring ctDNA levels during therapy with checkpoint inhibitors can inform on treatment response in real-time. Importantly, the mutant allele concentration rose either prior to or at progression for all patients, who experienced disease progression during the ctDNA monitoring period. This is in agreement with several other studies, indicating that evaluation of ctDNA during therapy may be a feasible supplement to conventional radiological imaging [12,14,16].

Nevertheless, one limitation using ctDNA analysis is the so-called non-shedders, which are patients where ctDNA cannot be detected. In our study, we were unable to detect ctDNA harboring mutations in *BRAF, NRAS,* or *TERT* in the baseline sample for seven (44%) of the patients. For four of these patients, this is most likely explained by the limited number of tumor-associated mutations that we assessed. However, the remaining three patients had BRAF mutations in their tumor, which we could not validate in plasma. Most of these patients had isolated metastases located in sites known to shed low levels of ctDNA, such as the brain and lungs [12,28,29,30]. Thus, alternative biomarkers would actually be needed to monitor these ctDNA negative patients. For this purpose, we chose to evaluate the association between baseline inflammatory cytokine levels and PFS. Here, we found that high baseline levels of MCP1 and TNFα were predictors of superior PFS in metastatic melanoma patients treated with checkpoint inhibitors. Combining MCP1 and TNFα into a cytokine score provided an even stronger predictor of PFS, suggesting that assessing multiple cytokines may be a very robust method of choice. Notably, based on our data, a cut-off value for high and low cytokine level could not be determined, and thus discrimination between high and low cytokine level solely applies to this specific cohort. Studies including larger cohorts are needed for further validation and determination of defined cytokine cut-off values. Of note, we found the Mesoscale TNFα assay to be more sensitive than the O-link TNFα assay. This explains why TNFα was detectable in samples analyzed by Mesoscale, but not O-link, emphasizing the importance of using sensitive methods in biomarker studies.

We hypothesize that the levels of the proinflammatory cytokines can be used as predictors of treatment response to checkpoint inhibitors owing to their important role during innate and adaptive immune activation. It has previously been shown that response to checkpoint inhibitor therapy is strongly associated with a high density of tumor-infiltrating lymphocytes (TILs) and an intra-tumoral IFNγ gene signature [31,34,42,43,44], suggesting that endogenous T cell activation is important for eliciting an effective response to checkpoint inhibitors. T cell activation is dependent on activated antigen presenting cells (APCs), and APCs are activated through the innate immune system. APC activation results in cytokine production and presentation of antigens by the APC to naïve T cells, which in turn become activated. Here, it is relevant to notice that dendritic cells secret several different cytokines upon activation, including MCP1 and TNFα. Therefore, we speculate that high levels of immune-related cytokines may reflect an activated innate and adaptive immune system within the tumor microenvironment. Hence, patients with high cytokine levels may be more primed to respond to checkpoint inhibitors.

ctDNA negativity and the cytokine score represent two distinct areas of biomarkers; ctDNA is tumor-specific and reflects the molecular composition of the tumor and possibly tumor burden, while cytokines are indicators of immune activation status. In the present study, we demonstrated that it was favorable for patients to have either undetectable ctDNA or a high cytokine score. However, we found no significant overlap between these two patient groups, indicating that ctDNA and cytokines can be used as independent biomarkers. One reason for this discrepancy can be the inability to detect ctDNA in patients with unknown mutations. Cytokines, on the other hand, are measurable if present in concentrations above detection limit. Thus, the cytokine score may synergize with ctDNA, yielding a strengthened predictive biomarker.

We are aware of the limitations of the study. The relatively small number of patients limits the extrapolation of the findings to the general clinical setup. Further validation of our findings in larger cohorts is therefore needed to validate results. However, it is worth noting that the association between ctDNA and PFS found in this study, is in agreement with a previous study and in general seems to be a well-established association [13]. Evaluation of cytokines as predictive biomarkers for responses to checkpoint inhibitors is an emerging field. A previous study found six cytokines (TRAIL, MCP1, IL2, TNFα, IL8, and IP10) to be associated with overall survival in two discovery cohorts, but were unable to verify the results in a validation cohort of 49 patients, emphasizing the need for validation [32]. The identification of cytokine-specific cut-off values using Receiver Operating Characteristic (ROC) analysis and validation of these in larger validation cohorts is important for verifying cytokines as predictive biomarkers.

## 5. Conclusions

In summary, in our cohort, undetectable ctDNA after 6–8 weeks of therapy and high baseline levels of MCP1 and TNFα are all individual predictors of superior PFS in metastatic melanoma patients treated with first-line checkpoint inhibitors. This current study suggests that ctDNA, MCP1, and TNFα can synergize and may in the future be utilized as a minimal invasive predictive biomarker for disease progression. Nonetheless, our results should be further validated in others melanoma cohorts, and preferably also explored in other cancer types treated with checkpoint inhibitors.

## Figures and Tables

**Figure 1 cancers-12-01414-f001:**
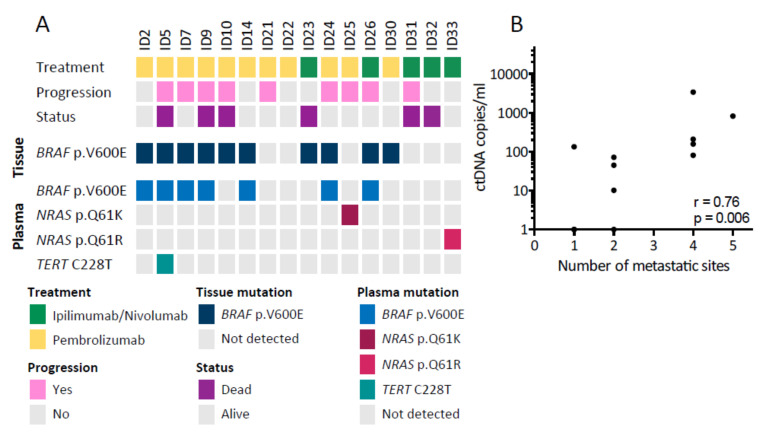
Detection of circulating tumor DNA (ctDNA) in baseline samples. (**A**) Overview of the patient cohort and the ctDNA mutations detected. The upper panel shows patient characteristics, while the lower panels show tissue and baseline ctDNA mutations detected. (**B**) Correlation between the baseline ctDNA level and number of metastatic sites (*n* = 12) analyzed with Spearman’s correlation coefficient (r).

**Figure 2 cancers-12-01414-f002:**
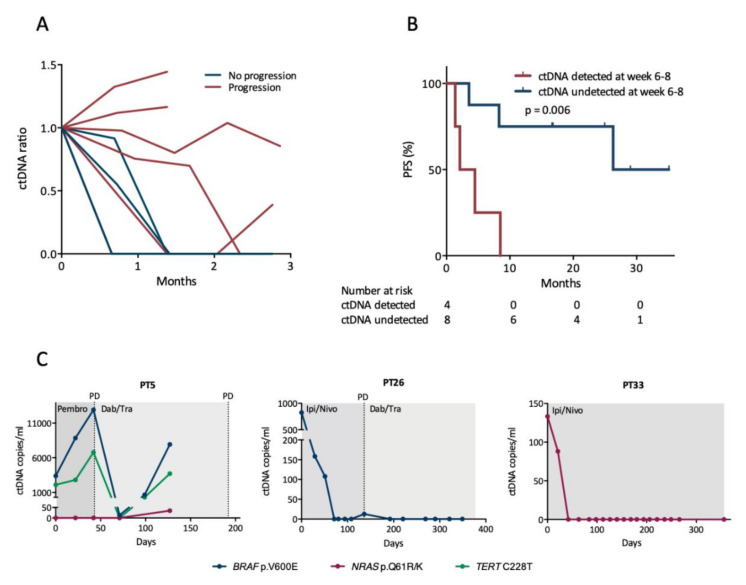
Early changes in circulating tumor DNA (ctDNA) levels. (**A**) Changes in ctDNA levels during the initial three months of first-line therapy. Lines discontinued before three months represent the last sample prior to or at disease progression. The ctDNA ratio reflects changes from the baseline sample. Only patients with ctDNA detected at baseline are represented (*n* = 6 with progression, *n* = 3 without progression). (**B**) Survival analysis for PFS according to whether ctDNA was detected at week 6–8 after therapy initiation. The difference between the groups was calculated using the log-rank test. (**C**) Examples of longitudinal monitoring of ctDNA levels during therapy. Time is depicted on the x-axis as days since treatment start. Abbreviations: Ipi/Nivo, ipilimumab/nivolumab; Dab/Tram, dabrafenib/trametinib; Pembro, pembrolizumab; PD, progressive disease.

**Figure 3 cancers-12-01414-f003:**
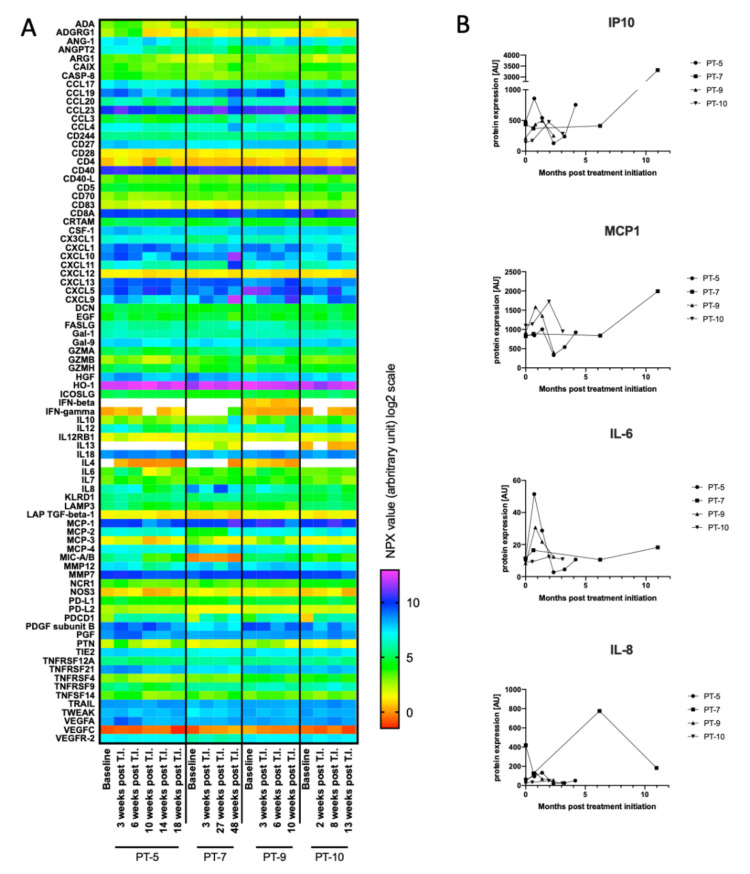
O-link screening for cytokines as potential biomarkers. (**A**) Baseline and follow-up blood samples from four patients were analyzed by O-link for the levels of 92 immuno-oncology proteins. All four patients were treated with first-line pembrolizumab. Normalized Protein eXpression (NPX) expression values (arbitrary units) are visualized on the heatmap. White color indicate undetectable cytokine level. The following cytokines were excluded as their concentrations were below detection limit: Fibroblast growth factor 2 (FGF2), Interleukin (IL) 1α, IL2, IL5, IL21, IL33, IL35, and tumor necrosis factor α (TNFα). (**B**) Graphs display plasma level changes for Interferon γ-induced protein 10 (IP10), monocyte chemoattractant protein 1 (MCP1), IL6, and IL8 over timer. A.U. = arbitrary units. T.I. = treatment initiation.

**Figure 4 cancers-12-01414-f004:**
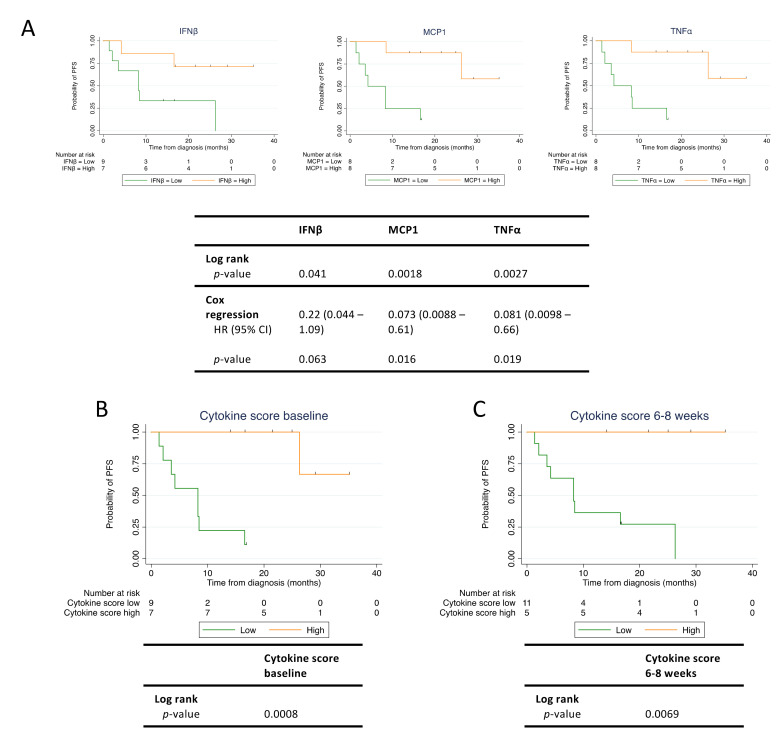
High cytokine levels predict longer progression-free survival (PFS) in checkpoint inhibitor treated patients. Baseline and follow-up blood samples were analyzed for levels of interferon β (IFNβ), monocyte chemoattractant protein 1 (MCP1), and tumor necrosis factor α (TNFα). (**A**) Survival analysis for PFS according to cytokine level in baseline blood sample. For each cytokine, the patients were dichotomized by the individual median cytokine concentration into a low and a high cytokine group. (**B**) Survival analysis for PFS according to cytokine score calculated from baseline MCP1 and TNFα levels. (**C**) Survival analysis for PFS according to cytokine score calculated from MCP1 and TNFα levels measured at week 6–8 after therapy initiation. Tick marks denote censored patients. (*n* = 16). HR; hazard ratio.

**Table 1 cancers-12-01414-t001:** Patient and disease characteristics.

Characteristic	Pembrolizumab	Ipilimumab/ Nivolumab	Total
No. of patients	11	5	16
Age-yr			
Median	54	62	57
Range	44–67	45–75	44–75
Sex-no. (%)			
Male	7 (63.6)	4 (80)	11 (69)
Female	4 (36.4)	1 (20)	5 (31)
Lactate dehydrogenase-no. (%)			
Normal	7 (63.6)	4 (80)	11 (69)
Elevated	4 (36.4)	1 (20)	5 (31)
BRAF V600 mutation, tissue-no. (%)	8 (72.7)	2 (40)	10 (62,5)
Tumour PD-L1 expression-no (%)			
<1%	1 (9)	5 (100)	6 (37.5)
≥1%	5 (45.5)	0 (0)	5 (31.3)
N/A	5 (45.5)	0 (0)	5 (31.3)
Tumour stage-no. (%)			
Stage III	1 (9)	0 (0)	1 (6)
Stage IV	10 (91)	5 (100)	15 (94)
Progression-no. (%)	7 (63.6)	2 (40)	9 (56.3)
MORS-no. (%)	3 (27.2)	3 (60)	6 (37.5)

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
