# Peer review of "Inflammatory Cytokines and ctDNA Are Biomarkers for Progression in Advanced-Stage Melanoma Patients Receiving Checkpoint Inhibitors"

_cancers, 2020, doi:10.3390/cancers12061414_

Round 1
Reviewer 1 Report
The author study ctDNA and cytokines as biomarkers for advanced-stage melanoma patients receiving checkpoint inhibitors. Unfortunately, the way the study is designed is not adeguate to the conclusion reported, and to the title.
In particular:
there is no comparison with patients receiving no therapy, or other therapies (targeted therapy). considering the lower number of patients enrolled it is crucial to understand the specificity of the results compared to other treatment.
there are no informations about stage III or IV patients, no correlation with the tumor burden at the starting point and the moment of measurement. thus, the diminished ctDNA could be the results of a reduced tumor burden, that is (per se) an indicator of better prognosis.
it is not clear if the level of cytokines detected is related to a better response to therapy, and not only to general PFS, and it is not associated to specific therapy.
The results of the experiments can't be associated to the IT benefit if there is no comparison with patients receiving other tyope of treatment or no treatment at all. for the same reason, those cytokines monitored could not be associated to the choice of IT versus targeted therapy, that was the main statement and rationale of the study.
Reviewer 2 Report
The authors prospectively examined 16 patients with metastatic melanoma treated with immune checkpoint inhibitors to evaluate whether circulating tumor DNA and/or inflammatory cytokines might have a role as potential biomarkers to predict treatment response and outcome. In their series, a decrease in the level of ct DNA at 6-8 weeks after treatment initiation predicts a longer progression free survival. Similarly, high levels of pre-treatment MCP1 and TNF alpha are associated with longer PFS than low levels of those cytokines.
This work is of high interest as biomarkers to predict outcome of patients are desperately needed in melanoma to predict which patients will respond or not to immune checkpoint inhibitors. This work must however be characterized as preliminary a best and is limited in the following manners:
- Most importantly, the number of patients included in this study is extremely small, which increases the risk that the findings of the study may not translate into a larger population of metastatic melanoma patients. It would be great if the authors could confirm their findings by increasing the cohort of patients, or (maybe even better) by validating their results in a separate cohort of patients. I suppose that the authors could at a minimum validate the cytokine (MCP1 and TNF alpha) in a separate population as the findings are based on pre-treatment levels only.
- As the ID numbers of the patients is not sequential and another identification tag is used in figure 3, it would be important to know whether a large number of patients available were excluded from the study (and if yes, why) and whether the patients included in the section 3.5 “initial characterization of cytokines…” belong to the same collection of 16 patients of the study. If yes, please use the same ID system in all studies, if no, please explain why. Similarly, please clarify whether the 4 patients of the validation group used in section 3.5 have been used in further analyses
- Discussion section. Please add a paragraph to highlight whether you estimate that ctDNA measurements will be predictive for responses only under immune checkpoint inhibitor therapy or also in other therapeutic approaches
Minor points:
Line 80: please change “considerablemuch” to “much” or “considerable”
Line 97: in every other part of the manuscript you describe that measurements were performed a 6-8 weeks. Please correct where appropriate
Reviewer 3 Report
The paper is well written, and the research is within an area that is currently garnering much interest within the cancer biomarker space. However, patient numbers are too small to draw a clinically worthwhile conclusion and the ctDNA findings are not new.
Major Comments
- Although the cytokine findings are potentially interesting and worth exploring further, the small numbers and lack of validation limits its use as a biomarker. Please provide validation with more patients, only looking at baseline levels using the U-plex 10 cytokine custom panel. Given the similar findings between baseline and Week 6-8 samples, longitudinal blood samples do not appear to be of great benefit when analysing cytokine levels.
- The CCR paper by Lim et al, which demonstrated that the CYTOX score could identify patients at risk of developing immune related adverse events, also reported on RECIST response and overall survival in a much larger cohort of patients. They also found that TNF alpha was associated with worse survival in patient who received combination immunotherapy, however, this was not validated in an independent cohort of patients. Please review this paper and include in your discussion, highlighting the importance of validation when reporting on biomarkers.
- What was the response rate of patients with high and low TNF alpha and MCP1? When you perform a ROC analysis, what was the AUC for both cytokines individually and combined in predicting response?
- For the O-Link analysis, TNF alpha was excluded due to undetectable levels in the input samples, whereas this cytokine is included in the U-plex panel. Please describe the fold change, limit of detection and how robust the TNF alpha results were when the U-plex panel was used. This point should also be included and discussed in the discussion.
- Some of your numbers are not consistent. In your statistics section, you mention that patients with BRAF V600E negative tumours and undetectable baseline ctDNA were excluded from all analysis of ctDNA. ctDNA was detectable in 9 patients, and BRAF V600 mutation was found in 11 patients (Table). But in the text, 12 patients had a mutation, 10 with BRAF V600 and 2 with NRAS Q61 mutation. Furthermore, longitudinal ctDNA analysis was performed on 12 patients, making your statement in the stats section incorrect.
Minor comments
- In figure 4a, IFN low = 9 and IFN high = 7. I suspect this is due to the median. What happens to your statistics if you dichotomise the patients who IFN low = 7 and IFN high = 9.
- Please include a multivariate analysis for PFS, including LDH, M stage, median tumour size and treatment type. These are well known clinical predictors of immunotherapy response.
- Please include interpatient variability of each of the 10 cytokines in the U-plex panel at baseline and demonstrate at median and interquartile range.
- Did you measure CRP? If so, what was the associated between CRP and TNF alpha/MCP1 levels.
Reviewer 4 Report
JG Pedersen and colleagues reported the results of biomarkers analyses in melanoma patients treated with immune checkpoint inhibitors (ICI) as first line.
The topic is of utmost importance since we do not have any reliable biomarkers of response to ICI in melanoma; the study is well written.
Major comments
The number of patients is low (n=16) and there some missing biological data for some of them (particularly ctDNA). As a consequence, the subgroups of analyses are extremely small and the analyses are limited to univariate analyses whose clinical relevance is questionable. Moreover, there is no paragraph about the limitations of the study in the discussion, to underline these.
The discussion, conclusion and abstract should be more balanced (this study suggests some interesting results but is not strong enough to “demonstrate”)
How do the authors explain that the LDH level is not associated with PFS in their study, while it is one of the strongest prognostic factors in melanoma? This result appears as a marker of the limitation of the extrapolability of the biomarker results in this small population, instead of “emphazising the continued need for better and more reliable biomarkers.” (L340)
- Why patients with BRAF wild type melanoma were excluded from the analysis of ctDNA? The number of patients is extremely low at study entry. Why patients were not screened for other driver mutations in tissue as NRAS at baseline? it would have increased the quantity of data.
- How do the authors explain that the baseline expression level of ctDNA is not correlated to PFS ?
- Significant results observed with ctDNA are presented at 6-8 weeks but samples were collected every 3 to 4 weeks; what were the early variation of ctDNA? Because predictive biomarkers are usually assessed at the beginning of treatment
- Considering the immune cytokine score, the authors identified that 4 cytokines had an impact on PFS (IFNb, TNF, MCP1 and IL1b) but they included only 2 of them in their final score. I understand that these 2 had the highest impact on PFS. But the 4 cytokines could have been included. I don't understand the reason of selection.
Minor comments
Method section:
- inclusion criteria L102-103: the first line treatment with ICI is not mentioned as an inclusion criterion, but I understand it was required. It should be clarified. Moreover, patients with BRAF mutated melanoma could have been included if they have been pretreated with BRAF and MEK inhibitors before their first-line ICI ?
- L106 Aarhus please add the country with brackets
- L133: why TERT was chosen as one of the 3 mutations detected by ddPCR ?
Results section:
- “Table 1” is misspelled in the text
- L188 for the patient who died of non cancer related adverse event: was it of an immune related adverse event? Could it be precised?
- L217: 130 plasma samples available: how many per patient? (median)
- L259 the literature analyzed to choose cytokines of interest should be cited
- L283: “the median PFS…” the results are difficult to visualize in the way they are presented. Is it possible rephrase? (ex: median PFS for high vs low IFNb, then idem for MCP1 …)
- in table 1: MORS : what is the signification of this acronym?
Round 2
Reviewer 1 Report
The author changed some part of the text.
CtDNA study is not specific for immunotherapy as reported in literature and as the author discussed in their response to me (should be included in the text) Cytokine work lack confirmation in patients not receiving IT but other therapies, so the title and the conclusion should be modified and report the correlation with IT benefit in this patients, but can't be pushed as criteria to select IT in this patients over targeted therapy.
Author Response
Notice: it has come to our attention that depending on how the manuscript file is viewed (with/without comments) the line numbers do change. Therefore, our reference to line numbers is based on the manuscript where comments are active.
We agree that ctDNA is not specific for immunotherapy only, but also applicable for targeted therapy. This is also mentioned in our discussion (L366-368).
The text has been modified to emphasize that the findings suggest that ctDNA and cytokines can be used to evaluate response to immunotherapy, but not to stratify patients to immunotherapy over other treatment type, e.g. targeted therapy (L17, 33, 49, 55, 90, 91, 417).
Other clarifications have been included:
For example, in the abstract, this sentence; “Yet, 40-60% of patients do not respond to therapy emphasizing that correct patient stratification to checkpoint inhibition is critical.” has been changed to; “. Yet, 40-60% of patients do not respond to therapy emphasizing the need for better predictive biomarkers for treatment response to immune checkpoint inhibitors.” to emphasize that the study focuses on identifying biomarkers that can predict treatment response to immune checkpoint inhibitors, but not on stratification of patients to specific treatment types.
Also, the paragraph “… and guide patients for checkpoint inhibitor therapy” has been deleted from the abstract.
In the introduction, we have changed the following sentence; “Thus, it remains of utmost importance to uncover better predictive biomarkers for the stratification of patients prior to therapy initiation, as well as for early assessment of genuine non-response.” to the following; “. Thus, it remains of utmost importance to uncover better predictive biomarkers for early assessment of response to immunotherapy.”
In the last section of the discussion we have deleted words from the sentence; “This current study was limited by the number of patients included, but suggests that ctDNA, MCP1, and TNFa can synergise and be utilised as a minimal invasive predictive biomarker for disease progression and guidance on modality”, since this cannot be concluded from the study, as correctly pointed out by the reviewer.
Finally, we accept to modify the title to the following “Inflammatory Cytokines and ctDNA Are Biomarkers for PFS in Advanced-Stage Melanoma Patients Re-ceiving Checkpoint Inhibitors”
Reviewer 2 Report
The concerns of the reviewer have been addressed and the publication can, in my opinion, be considered for publication.
Please ensure that the following minor modifications are performed:
L98: replace “no presence of” by “absence of”
L395: We are aware of the limitations of the study:
L396: of the findings to the general clinical setup
L397: change “to verify” to “to validate”
L400: is an emerging
L404-405: consider deleting the sentence: Despite the lack of a validation cohort for this study, we believe these findings are still important.
L406 add: In summary, in our cohort, undetectable
Author Response
Notice: it has come to our attention that depending on how the manuscript file is viewed (with/without comments) the line numbers do change. Therefore, our reference to line numbers is based on the manuscript where comments are active.
The concerns of the reviewer have been addressed and the publication can, in my opinion, be considered for publication.
Please ensure that the following minor modifications are performed:
L98: replace “no presence of” by “absence of”
Corrected
L395: We are aware of the limitations of the study:
Corrected
L396: of the findings to the general clinical setup
Corrected
L397: change “to verify” to “to validate”
Corrected
L400: is an emerging
Corrected
L404-405: consider deleting the sentence: Despite the lack of a validation cohort for this study, we believe these findings are still important.
Sentence has been deleted
L406 add: In summary, in our cohort, undetectable
Corrected
Reviewer 3 Report
Despite addressing the limitations in the conclusions, the authors still have the issue of very small patient numbers and the lack of validation.
The results do not support the conclusion, and the cytokine results were not supported by already published studies which looked at much higher numbers of patients.
The authors commented that a validation cohort will take 27 months, but have they reached out to other institutions for collaboration etc. to address this. This will only require 15-20 patients, and small volumes of plasma.
Author Response
Despite addressing the limitations in the conclusions, the authors still have the issue of very small patient numbers and the lack of validation.
The results do not support the conclusion, and the cytokine results were not supported by already published studies which looked at much higher numbers of patients.
With respect to the reviewer comments, we do not believe our conclusions are to extensive. We attempt to balance our results and conclusion proviso the size of the cohort.
The text and conclusions have been modified (L17, 33, 49, 85, 90, 91, 364, 365, 411).
In the abstract, the following sentence; “Yet, 40-60% of patients do not respond to therapy emphasizing that correct patient stratification to checkpoint inhibition is critical.” has been changed to; “. Yet, 40-60% of patients do not respond to therapy emphasizing the need for better predictive biomarkers for treatment response to immune checkpoint inhibitors.”
In the abstract, the conclusion has been moderated to include “suggest” in L30; “Conclusions: These findings suggest that the levels of ctDNA, MCP1, and TNFa in baseline and early follow-up samples can predict disease progression in metastatic melanoma patients treated with checkpoint inhibitors.”. Additionally, we have deleted; “and guide patients for checkpoint inhibitor therapy” from the abstract.
In the last section of the discussion we have deleted wordings from the sentence; “This current study was limited by the number of patients included, but suggests that ctDNA, MCP1, and TNFa can synergise and be utilised as a minimal invasive predictive biomarker for disease progression and guidance on modality”, since this cannot be concluded from the study.
In regard to already published studies, to our knowledge there are currently no such studies that have demonstrated and validated an association between PFS and MCP1 or TNF-alpha in cancer patients treated with immunotherapy.
The authors commented that a validation cohort will take 27 months, but have they reached out to other institutions for collaboration etc. to address this. This will only require 15-20 patients, and small volumes of plasma.
Overall, we do not disagree with the reviewer that a validation cohort would be interesting. However, such cohort would need to be designed in a similar manner as ours, including blood sampling at same timepoints, collection and storage of plasma samples, inclusion criteria etc. Thus we find this request to be out of scope of the current hypothesis creative manuscript.
Reviewer 4 Report
The authors provided a revised version of their article, but the remarks were only partially considered.
They did not respond to 2 comments (points 13 and 14; point 9 incomplete) in their cover letter. Modifications according to point 9 don’t seem to be included in the text.
Please refer to the initial batch of comments.
A more substantially revised version and a more complete response should be submitted for consideration.
Author Response
The authors provided a revised version of their article, but the remarks were only partially considered.
We regret that the reviewer did not find our answers satisfactory. We believe that the line numbering of the manuscript changes dependent on whether comments are present or not, and thus it may be difficult to find our corrections and responds to the reviewer. Please see below a more thoroughly description of modifications and where to find them in the manuscript.
They did not respond to 2 comments (points 13 and 14; point 9 incomplete) in their cover letter. Modifications according to point 9 don’t seem to be included in the text.
Please refer to the initial batch of comments.
A more substantially revised version and a more complete response should be submitted for consideration.
(Points from first rebuttal letter)
POINT 9: L133: why TERT was chosen as one of the 3 mutations detected by ddPCR ?
Respond 9: TERT was chosen as this is one of the frequent mutations in melanoma, found in 29-43% of melanomas (Griewank G.K et al 2014 JNCI; Vinagre J. et al 2013 Nature Communication). Owing to technical limitations with the TERT assay, we were only able to detect one type of TERT mutations.
Update: this is included in the method section section 2.5, forth line- “The TERT assay was designed and validated in-house. Due to technical limitations with the TERT assay, only the C228T mutation was approved for this study.”
And in the result section“3.2. Detection of ctDNA at Baseline” line 1 and 2.
cfDNA was analysed for BRAF p.V600E, NRAS p.Q61K/p.Q61R, and TERT C228T mutations, as these are frequentlty found in melanoma patients 24–27. ctDNA was detected by ddPCR at base-line in 9/16 patients (56 %). “
POINT 13: L259 the literature analyzed to choose cytokines of interest should be cited
Respond 13: references have been added to the manuscript. We searched for articles that had previously analysed cytokines as biomarkers for treatment response to checkpoint inhibitors (Yamazaki, N et al 2017 Cancer Science; Lim, S Y et al 2018 Clinical Cancer Research). Additionally, we also searched for cytokines involved in generating pro- and anti-tumor immune responses (Karachaliou, N et al 2018 Therapeutic advances in medical oncology; Demaria, O et al 2019 Nature; Ayers, M et al 2017 Journal of clinical investigation).
Please find them in section 3.5, third last line, now as reference 21, 31-34.
POINT 14: L283: “the median PFS…” the results are difficult to visualize in the way they are presented. Is it possible rephrase? (ex: median PFS for high vs low IFNb, then idem for MCP1 …)
Respond 14: The sentence is modified to; “The median PFS was; 8.2 months for low IFNb and not reached for high IFNb; 4.2 months for low MCP1 and not reached for high MCP1; 4.2 months for low TNFa and not reached for high TNFa.”.
Due to text edits in the manuscript this can now be read section 3.6 last three lines.
Round 3
Reviewer 4 Report
The authors provided a more complete revision of their article. They improved the presentation of the results and the discussion and conclusion are more balanced.
Last comment:
- I don’t understand why the title was modified; the abbreviation “PFS” should be avoided in a title, and could be replaced by “of progression”
Author Response
The title was changed to balance the conclusion, as requested by reviewer 1.
We agree that abbreviations should be avoided in titles, and PFS has now been replace by progression. The title is now: “Inflammatory Cytokines and ctDNA Are Biomarkers for progression in Advanced-Stage Melanoma Patients Receiving Checkpoint Inhibitors”.